# An Adaptive Controller Based on Interconnection and Damping Assignment Passivity-Based Control for Underactuated Mechanical Systems: Application to the Ball and Beam System

**Xiaoping Liu [1],\*, Huaizhi Shao [2], Cungen Liu [2], Ning Li [2], Xinpeng Guo [3], Fei Zheng [4] and Lijun Sun [5]**

1 Faculty of Engineering, Lakehead University, Thunder Bay, ON P7B 5E1, Canada
2 School of Information and Electrical Engineering, Shandong Jianzhu University, Jinan 250101, China; hzshao22@163.com (H.S.); littleeggs@sdjzu.edu.cn (C.L.); nli2falw@163.com (N.L.)
3 Shandong Telian Information Technology Co., Ltd., Jinan 250101, China; guoxp@itelian.cn
4 Shandong Marvelous Intelligent Technology Co., Ltd., Jinan 250000, China; zflyimage@163.com
5 College of Information Science and Engineering, Henan University of Technology, Zhengzhou 450001, China; ljsun@haut.edu.cn
\* Correspondence: xliu2@lakeheadu.ca

**Abstract:** In this paper, an adaptive technology and the interconnection and damping assignment passivity-based control method are combined to solve the stabilization problem for underactuated mechanical systems with uncertainties (including matched and unmatched). These uncertainties include unknown friction coefficients and unknown terms in kinetic energy and potential energy. A novel adaptive interconnection and damping assignment passivity-based control scheme is proposed and an adaptive stabilization controller is designed to make the closed-loop system locally stable. Verification is conducted on the ball and beam system. The locally asymptotic stability is demonstrated using the LaSalle's invariance principle and approximate linearization. The effectiveness of the proposed control law is verified through numerical simulations.

**Keywords:** underactuated mechanical systems; adaptive control; IDA-PBC; the ball and beam system



## 1. Introduction

The port-controlled Hamiltonian (PCH) model, which is regarded as another alternative model for the Euler–Lagrange model, is widely used to describe dynamic equations for nonlinear systems. The system described by the PCH structure has many advantages: a number of natural physical systems are covered, and significant structural properties are preserved. The independent control quantity of the system is less than its degree of freedom to be controlled, and a system with this property is called an underactuated system [1], and its dynamics are usually nonlinear, which is of significant difficulty to control. One of the effective technologies used to control underactuated physical systems is interconnection and damping assignment passivity-based control (IDA-PBC) [2], which has been resoundingly used to solve the stabilization problems of various underactuated systems described by the PCH framework. Moreover, this technology has been extensively used in induction machines [3], power converters [4], flexible spacecrafts [5], aircrafts [6] and so on.

However, one of the main shortcomings of the IDA-PBC method is that a set of partial differential equations (PDEs) need to be solved. In order to simplify this problem, outstanding contributions have been made by a large number of researchers. For instance, in [7], by parameterizing the expected inertia matrix, the potential PDE was enormously simplified, and this approach was extended to separable and nonseparable PCH systems. In order to ensure the solvability of PDEs, some conditions were added to the expected structure matrices, $J_d$ and $R_d$, which were allowed to depend on the control input [8]. The good performance of this technique was demonstrated by the well-known boost power

converter. In addition, some constructive solutions were also proposed to simplify PDEs of underactuated mechanical systems (UMSs) in [6,9–11].

Many theoretical extensions and practical research on the IDA-PBC approach have been reported in the literature. In [12], two design methods of IDA-PBC were proposed in view of the existence of physical damping in the Hamiltonian frame. By combining the data sampling method with IDA-PBC, a sampling data controller [13] was designed, and the target dynamics was stabilized to the equilibrium point. In order to tolerate the limitation of actuator faults, the IDA-PBC method with fault tolerance was improved in [14], and a high-gain adaptive IDA-PBC scheme was proposed. The effectiveness of the improved control law was verified by the experiment of a hexarotor UAV.

Furthermore, the robustness of IDA-PBC strategies to disturbances has also been a hot topic in recent years. As reported in [15], an outer-loop controller was designed to solve the matched disturbance suppression problem of UMSs. In [16,17], a new IDA-PBC law was constructed by combining a model reference adaptive control method with IDA-PBC, which could more effectively compensate for disturbances compared to the standard IDA-PBC in [2]. In [18,19], a method of adding integral effects to IDA-PBC was presented for a kind of UMS with constant disturbances. In order to solve the problem of matched and unmatched disturbance suppression, specific coordinate changes were added to the damping term in [20]. Ref. [21] proposed a novel IDA-PBC scheme for a quadrotor aircraft based on a filter observer that could deal with the output measurement of large noise signals and uncertainties in the translation and rotation dynamics. The simulation results of the quadrotor aircraft showed that the sensitivity of the noise measurement was significantly improved and the steady-state error was reduced. As far as UMSs are concerned, external interference is also abundant, which cannot be ignored during system modeling. In [22], the IDA-PBC approach was applied to an inertial wheel inverted pendulum, and the results showed that it had good robustness to external interference. Due to the change in parachute mass and the existence of wind, ref. [23] proposed two control algorithms for an unmanned powered parachute aircraft based on PBC. The numerical simulation showed that the IDA-PBC algorithm based on the Hamiltonian function was unaffected by the parachute mass change and wind speed. In [24], a novel robust state error IDA-PBC algorithm was developed for unmanned surface vessels. Through the combination of a reduced-order extended state observer, the state error IDA-PBC technology and the auxiliary dynamic system, the tracking performance was improved and the system's energy consumption was reduced. Simulations showed that the proposed control strategy ensured the asymptotic stability of the system's signals. Considering that the inertia matrix depends on non-actuated coordinates for underactuated systems, an integral effect with specific coordinate transformations was added to the outer-loop of the IDA-PBC scheme in [25]. The designed control scheme was applied to a UAV, which proved its effectiveness. In addition, the influence of viscous friction was studied by using the controlled Lagrangian method [26], and the closed-loop system was more stable. In [27], the IDA-PBC strategy was used to analyze continuous friction.

A well-known UMS is the ball and beam system. It is composed of a beam that can rotate along the horizontal axis and a ball that lies on the beam. The control goal is to make the ball reach the equilibrium position from any initial position at any initial speed by applying torque to the beam. There are many existing modeling and control methods for the ball and beam system. In both [28,29], the Euler–Lagrange method was used to model a ball and beam system. In [29], the equations of the ball and beam system in [28] were scaled according to time and torque. In the past few years, more and more new control methods have been used to control a ball and beam system, such as fuzzy logic, neural networks, robust control and backstepping [30–33]. A comparative study was conducted for models of ball and beam systems in [34]. Moreover, the $\lambda$-method matching control law was first applied to a ball and beam system, and the experimental results showed that the theoretical prediction was consistent with the experimental results in [35]. Ref. [36]

proposed a new control scheme to eliminate the influence of matched and mismatched disturbances, which combined time scaling with the redesign of Lyapunov.

Considering the above situation, an IDA-PBC scheme based on an adaptive method is proposed in this paper in view of the unknown frictions in UMSs and uncertainties in the modeling process, which are better compensated. Only the matched input disturbances were considered in [15,37], and the only external frictions of the system were compensated in [22,38]. Finally, the uncertainties in friction and potential energy were handled, respectively, in [39]. Compared with the above, the uncertainties in external frictions, the inertia matrix, $M$, and the potential energy, $V$, are estimated adaptively in this paper, which expands the research scope.

The main contributions can be summarized as follows.

(1) An adaptive controller is designed for a UMS with unknown parameters in the inertia matrix, potential function and friction coefficients.

(2) The estimate values of the unknown terms are placed in the damping injection controller, $u_{di}$, instead of the energy shaping controller, $u_{es}$, which simplifies the solution of the partial differential equations.

(3) By using LaSalle's invariance principle and approximate linearization, the locally asymptotic stability of the state of the ball and beam system is achieved.

The rest of the paper is organized as follows: In Section 2, the design steps of IDA-PBC are briefly reviewed, and the problems to be solved are formulated. A new adaptive controller is proposed, and a stability analysis is given in Section 3. In Section 4, the new control scheme is applied to the ball and beam system, and numerical simulation results are provided. Finally, a summary is presented in Section 5.

## 2. Problem Statement

In this section, the standard IDA-PBC method [2] for UMSs is briefly looked at. The various possible uncertainties are discussed, and the PCH system with uncertainties is presented.

### 2.1. Review of IDA-PBC Design

Consider a mechanical system defined by

$$M(q)\ddot{q} + \left( M(q) - \frac{1}{2}\frac{\partial \dot{q}^T M(q)}{\partial q} \right)\dot{q} + \nabla_q V(q) = G(q)u, \tag{1}$$

where $M(q) = M^T(q) > 0$ is the inertia matrix, $V(q)$ is the potential energy function, $q \in \mathbb{R}^n$ is the generalized position and $u \in \mathbb{R}^m$, $m \leq n$ is the control input. The matrix $G(q) \in \mathbb{R}^{n \times m}$ is an input matrix. The system is called fully actuated when $m = n$ and $rank(G) = m = n$, whereas it is called underactuated when $m < n$ and $rank(G) = m < n$. $\nabla_q V$ is the gradient of $V(q)$, i.e., $\nabla_q V(q) = \frac{\partial V(q)}{\partial q}$.

The Hamiltonian function, $H$, which is defined as the sum of the kinetic energy and the potential energy, is the total energy of the system. It can be written as

$$H(q, p) = \frac{1}{2}p^T M^{-1}(q)p + V(q), \tag{2}$$

where $p = M\dot{q}$ is momenta. Then, the dynamic Equation (1) can be represented in the following PCH form

$$\begin{bmatrix} \dot{q} \\ \dot{p} \end{bmatrix} = \begin{bmatrix} 0 & I_n \\ -I_n & 0 \end{bmatrix} \begin{bmatrix} \nabla_q H \\ \nabla_p H \end{bmatrix} + \begin{bmatrix} 0 \\ G(q) \end{bmatrix} u$$

$$y = G^T(q)\nabla_p H, \tag{3}$$

where $y \in \mathbb{R}^m$ is the output. $I_n$ represents the $n \times n$ identity matrix.

The IDA-PBC method is composed of two parts, namely energy shaping and damping injection, i.e.,

(1) Energy Shaping: The state feedback controller, $u_{es}$, should be designed so that the closed-loop system takes the following form

$$
\begin{bmatrix} 0 & I_n \\ -I_n & 0 \end{bmatrix} \begin{bmatrix} \nabla_q H \\ \nabla_p H \end{bmatrix} + \begin{bmatrix} 0 \\ G(q) \end{bmatrix} u_{es} = \begin{bmatrix} 0 & M^{-1}M_d \\ -M_d M^{-1} & J_2(q,p) \end{bmatrix} \begin{bmatrix} \nabla_q H_d \\ \nabla_p H_d \end{bmatrix}. \tag{4}
$$

Here, $J_2 = -J_2^T = \begin{bmatrix} 0 & j \\ -j & 0 \end{bmatrix}$ is a free parameter and $H_d$ is the desired Hamiltonian function, which is defined by

$$
H_d(q,p) = \frac{1}{2} p^T M_d^{-1}(q) p + V_d(q), \tag{5}
$$

with $M_d(q) = M_d^T(q) > 0$ and $V_d(q)$ representing the desired inertia matrix and the desired potential energy, respectively. It is assumed that there is an isolated minimum at the desired equilibrium point $q_*$, i.e.,

$$
q_* = \arg\min H_d(q) = \arg\min V_d(q). \tag{6}
$$

This is true if the conditions $\nabla_q V_d(q_*) = 0$ and $\nabla_q^2 V_d(q_*) > 0$ are satisfied.

It can be easily verified that the first line of Equation (4) is satisfied. The second line of Equation (4) can be written as

$$
-\nabla_q H + G(q) u_{es} = J_2(q,p) \nabla_p H_d - M_d M^{-1} \nabla_q H_d,
$$

which is equivalent to

$$
G(q) u_{es} = \nabla_q H + J_2(q,p) \nabla_p H_d - M_d M^{-1} \nabla_q H_d. \tag{7}
$$

Let $G^\perp$ represent a full rank left annihilator of $G$, i.e., $G^\perp G = 0$. As a result, multiplying (7) by $G^\perp$ from the left-hand side gives

$$
G^\perp \left\{ \nabla_q H + J_2 \nabla_p H_d - M_d M^{-1} \nabla_q H_d \right\} = 0. \tag{8}
$$

The PDE (8) can be equivalently written as the following two PDEs:

$$
G^\perp \left\{ \nabla_q \left( p^T M^{-1} p \right) - M_d M^{-1} \nabla_q \left( p^T M_d^{-1} p \right) + 2 J_2 \nabla_p H_d \right\} = 0, \tag{9}
$$

$$
G^\perp \left\{ \nabla_q V - M_d M^{-1} \nabla_q V_d \right\} = 0. \tag{10}
$$

The energy shaping control law, $u_{es}$, can be determined as

$$
u_{es} = \left( G^T G \right)^{-1} G^T \left( \nabla_q H - M_d M^{-1} \nabla_q H_d + J_2 M_d^{-1} p \right). \tag{11}
$$

by solving (9) for $J_2$ and $M_d$, and (10) for $V_d$.

(2) Damping Injection: The object is to design a damping injection controller,

$$
u_{di} = -K_v G^T \nabla_p H_d, \tag{12}
$$

where $K_v = K_v^T > 0$ is a parameter matrix.

By using the controller

$$
u = u_{es} + u_{di}, \tag{13}
$$

the given PCH system (3) is made to have the following expected PCH dynamics:

$$\begin{bmatrix} \dot{q} \\ \dot{p} \end{bmatrix} = [J_d(q,p) - R_d(q,p)]\begin{bmatrix} \nabla_q H_d \\ \nabla_p H_d \end{bmatrix} \tag{14}$$

$$y_d = G^T(q)\nabla_p H_d, \tag{15}$$

where $J_d$ and $R_d$ are the redistributed expected interconnection and damping matrices, defined by

$$J_d = -J_d^T = \begin{bmatrix} 0 & M^{-1}M_d \\ -M_d M^{-1} & J_2 \end{bmatrix},$$

$$R_d = R_d^T = \begin{bmatrix} 0 & 0 \\ 0 & GK_vG^T \end{bmatrix} \geq 0.$$

The desired Hamiltonian function (5) is considered as a candidate Lyapunov function. Its derivative is

$$\begin{aligned} \dot{H}_d &= (\nabla_p H_d)^T \dot{p} + (\nabla_q H_d)^T \dot{q} \\ &= p^T M_d^{-1}(-\nabla_q H + G(u_{es} + u_{di})) + (\nabla_q H_d)^T M^{-1} p \\ &= p^T M_d^{-1} J_2 M_d^{-1} p - p^T M_d^{-1} GK_v G^T \nabla_p H_d \\ &= -(\nabla_p H_d)^T GK_v G^T \nabla_p H_d \leq 0. \end{aligned}$$

Thus, $(q_*, 0)$ is a stable equilibrium point of systems (14) and (15). In addition, if the zero-state of the closed-loop systems (14) and (15) can be detected from their outputs (15), then the equilibrium point $(q_*, 0)$ is asymptotically stable.

*2.2. Possible Uncertainties*

It is usually assumed that all the parameters of the system are known when using the IDA-PBC method. However, uncertainties exist inevitably in reality, which may lead to poor control performance. In the PCH system (3), uncertainties might occur in the Hamiltonian function $H$. In addition, frictions $f$ exist in almost all mechanical systems.

Consider the following dynamic model:

$$M\ddot{q} + \Delta M\ddot{q} + \left(M - \frac{1}{2}\frac{\partial \dot{q}^T M}{\partial q}\right)\dot{q} + \left(\Delta M - \frac{1}{2}\frac{\partial \dot{q}^T \Delta M}{\partial q}\right)\dot{q} + (\nabla_q V + \nabla_q \Delta V) = Gu^* - f, \tag{16}$$

where $\Delta M$ and $\Delta V$ denote unknown terms in $M$ and $V$, respectively, and $f = diag\{\dot{q}_1, \dot{q}_2, \cdots, \dot{q}_n\}\theta$ represents frictions, with $\theta = [\theta_1, \theta_2, \cdots, \theta_n]^T$ being frictional coefficients. Define $H$ as (2) and $p = M\dot{q}$. Then, it can be verified that (16) can be changed to the following PCH form:

$$\begin{bmatrix} \dot{q} \\ \dot{p} \end{bmatrix} = \begin{bmatrix} 0 & I_n \\ -I_n & 0 \end{bmatrix}\begin{bmatrix} \nabla_q H \\ \nabla_p H \end{bmatrix} + \begin{bmatrix} 0 \\ G \end{bmatrix}u^* - \begin{bmatrix} 0 \\ \Phi^T \beta \end{bmatrix}, \tag{17}$$

where $\Phi^T \beta = \Delta M\ddot{q} + \left(\Delta M - \frac{1}{2}\frac{\partial \dot{q}^T \Delta M}{\partial q}\right)\dot{q} + \nabla_q(\Delta V) + f$ represents a parameterization of the uncertainties, with $\beta = [\beta_1, \beta_2, \cdots, \beta_n]^T$ being a vector of unknown constant parameters.

## 3. Controller Design and Stability Analysis

In this section, an adaptive controller is designed to compensate for uncertainties, which was discussed in the previous section.

**Theorem 1.** *For the PCH system (17) with an unknown term, $\Phi^T\beta$, consider the closed-loop system*

$$
\begin{bmatrix} \dot{q} \\ \dot{p} \end{bmatrix} = \begin{bmatrix} 0 & M^{-1}M_d \\ -M_dM^{-1} & J_2 \end{bmatrix} \begin{bmatrix} \nabla_q H_d^* \\ \nabla_p H_d^* \end{bmatrix} - \begin{bmatrix} 0 \\ \Phi^T\tilde{\beta} \end{bmatrix}
\tag{18}
$$

$$
y_d^* = G^T\nabla_p H_d^*.
\tag{19}
$$

*This corresponds to the adaptive controller*

$$
u^* = u_{es}^* + u_{di}^*,
\tag{20}
$$

$$
\dot{\hat{\beta}} = -\gamma\Phi\nabla_p H_d^*,
\tag{21}
$$

*with*

$$
u_{es}^* = \left(G^TG\right)^{-1}G^T\left(\nabla_q H - M_dM^{-1}\nabla_q H_d^* + J_2\nabla_p H_d^*\right).
\tag{22}
$$

$$
u_{di}^* = \left(G^TG\right)^{-1}G^T\left(-GK_vG^T\nabla_p H_d + \Phi^T\hat{\beta}\right),
\tag{23}
$$

*and the desired Hamiltonian function:*

$$
H_d^* = \frac{1}{2}p^T M_d^{-1}p + V_d + \frac{1}{2\gamma}\tilde{\beta}^T\tilde{\beta}.
\tag{24}
$$

*Here, $\tilde{\beta} = \beta - \hat{\beta}$, $\hat{\beta}$ is the estimated value of the unknown constant parameter $\beta$ and $\gamma = diag\{\gamma_1, \gamma_2, \cdots, \gamma_n\} > 0$ is the controller parameters. Assume that the detectability condition of the output (19) is satisfied. Then, $(q_*, 0) = (0, 0)$ is a locally asymptotically stable equilibrium point of the closed-loop systems (18) and (19), and $\tilde{\beta}$ is bounded.*

**Proof of Theorem 1.** The energy shaping controller, $u_{es}^*$, should be constructed so that

$$
\begin{bmatrix} 0 & I_n \\ -I_n & 0 \end{bmatrix} \begin{bmatrix} \nabla_q H \\ \nabla_p H \end{bmatrix} + \begin{bmatrix} 0 \\ G \end{bmatrix}u_{es}^* - \begin{bmatrix} 0 \\ \Phi^T\beta \end{bmatrix}
$$
$$
= \begin{bmatrix} 0 & M^{-1}M_d \\ -M_dM^{-1} & J_2 \end{bmatrix} \begin{bmatrix} \nabla_q H_d^* \\ \nabla_p H_d^* \end{bmatrix} - \begin{bmatrix} 0 \\ \Phi^T\tilde{\beta} \end{bmatrix}.
\tag{25}
$$

The first line of the equation is satisfied automatically, but the second line becomes

$$
Gu_{es}^* = \nabla_q H - M_dM^{-1}\nabla_q H_d + J_2\nabla_p H_d^*.
\tag{26}
$$

By premultiplying $G^\perp$, it follows from the above equation that

$$
G^\perp\left\{\nabla_q H + J_2\nabla_p H_d^* - M_dM^{-1}\nabla_q H_d\right\} = 0.
\tag{27}
$$

Equation (27) can be divided into the following two PDEs:

$$
G^\perp\left\{\nabla_q\left(p^T M^{-1}p\right) - M_dM^{-1}\nabla_q\left(p^T M_d^{-1}p\right) + 2J_2\nabla_p H_d^*\right\} = 0,
\tag{28}
$$

$$
G^\perp\left\{\nabla_q V - M_dM^{-1}\nabla_q V_d\right\} = 0.
\tag{29}
$$

Premultiplying (26) by $G^T$ and solving for $u_{es}^*$ produces

$$
u_{es}^* = \left(G^TG\right)^{-1}G^T\left(\nabla_q H - M_dM^{-1}\nabla_q H_d^* + J_2\nabla_p H_d^*\right).
$$

The derivative of (24) along the trajectories of (18) and (19) is

$$
\begin{aligned}
\dot{H}_d^* &= p^T M_d^{-1} \dot{p} + (\nabla_q V_d)^T \dot{q} - \frac{1}{\gamma} \tilde{\beta}^T \dot{\hat{\beta}} \\
&= p^T M_d^{-1} \left( -\nabla_q H + G(u_{es}^* + u_{di}^*) - \Phi^T \beta \right) + (\nabla_q V_d)^T \nabla_p H - \frac{1}{\gamma} \tilde{\beta}^T \dot{\hat{\beta}} \\
&= p^T M_d^{-1} \left( -M_d M^{-1} \nabla_q H_d^* + J_2 \nabla_p H_d^* + G u_{di}^* - \Phi^T \beta \right) + (\nabla_q V_d)^T M^{-1} p - \frac{1}{\gamma} \tilde{\beta}^T \dot{\hat{\beta}} \\
&= -\left( p^T M^{-1} \nabla_q H_d^* \right)^T + p^T M_d^{-1} J_2 \nabla_p H_d^* - p^T M_d^{-1} \Phi^T \beta + p^T M_d^{-1} G u_{di}^* \\
&\quad + (\nabla_q V_d)^T M^{-1} p - \frac{1}{\gamma} \tilde{\beta}^T \dot{\hat{\beta}} \\
&= -\frac{1}{\gamma} \tilde{\beta}^T \dot{\hat{\beta}} + p^T M_d^{-1} G u_{di}^* - p^T M_d^{-1} \Phi^T \beta.
\end{aligned}
\tag{30}
$$

By substituting (21) and (23) into (30), the following inequality is obtained:

$$
\dot{H}_d^* = -p^T M_d^{-1} G K_v G^T M_d^{-1} p \leq 0.
\tag{31}
$$

It follows from (30) that the desired equilibrium point, $(q_*, 0, 0)$, is stable. Furthermore, since the output (19) is locally zero-state detectable, the local asymptotic stability of the state is guaranteed. $\square$

## 4. Example: The Ball and Beam System

In this section, the well-known ball and beam system [28] is considered to have uncertainties, including uncertainties in the friction coefficients and uncertainties in the Hamiltonian function, $H$.

### 4.1. System Model

As shown in Figure 1, the dynamic behavior of the ball and beam system [28] is described as

$$
\begin{aligned}
\frac{c_2}{R^2} \ddot{q}_1 + M_b g \sin(q_2) - M_b q_1 \dot{q}_2^2 + \beta_1 \dot{q}_1 &= 0 \\
c_1 \ddot{q}_2 + 2 M_b q_1 \dot{q}_1 \dot{q}_2 + M_b g q_1 \cos(q_2) + \beta_2 \dot{q}_2 &= u,
\end{aligned}
\tag{32}
$$

where $c_1 = M_b q_1^2 + J + J_b$, $c_2 = M_b R^2 + J_b$, $c_3 = \frac{c_2}{c_1}$ and $M_b$, $R$ are the mass and the radius of the ball, respectively, $q_1$ and $q_2$ are the position of the ball and the angle of the beam, respectively, $J_b$ and $J$ are the moment of inertia of the ball and the beam, respectively, $u$ is the torque applied to the beam and $\beta_1 and \beta_2$ are the friction coefficients. Due to the ball being always maintained on the beam, the angle of the bar, $q_2$, is assumed to be $q_2 \in (-180°, 180°)$. Additionally, from (32), the inertia matrix, $M$, and potential energy function, $V$, are attained:

$$
M(q) = \begin{bmatrix} \frac{c_2}{R^2} & 0 \\ 0 & c_1 \end{bmatrix},
\tag{33}
$$

$$
V(q) = M_b g q_1 \sin(q_2),
\tag{34}
$$

and $G = \begin{bmatrix} 0 & 1 \end{bmatrix}^T$.

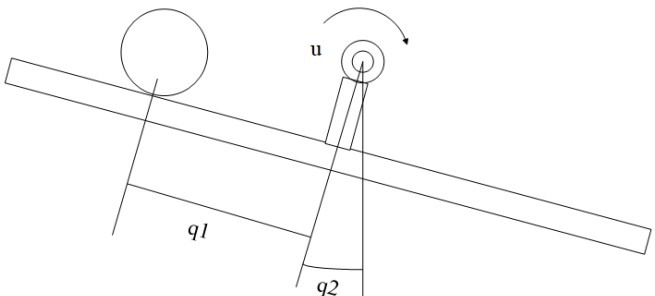

**Figure 1.** The ball and beam system.

The friction coefficients, the moment of inertia of the beam, $J$, in the inertia matrix and the gravitational constant, $g$, in the potential energy function are assumed to be unknown, which includes both matched and unmatched coefficients. Therefore,

$$
\begin{aligned}
\Phi^T \beta &= \Delta M \ddot{q} + \left( \Delta M - \frac{1}{2} \frac{\partial \dot{q}^T \Delta M}{\partial q} \right) \dot{q} + \nabla_q (\Delta V) + f \\
&= \begin{bmatrix} M_b \beta_3 \sin(q_2) + \beta_1 \dot{q}_1 \\ \beta_4 (\ddot{q}_2 + \dot{q}_2) + M_b \beta_3 q_1 \cos(q_2) + \beta_2 \dot{q}_2 \end{bmatrix}.
\end{aligned}
\tag{35}
$$

Here, $\Delta M = \begin{bmatrix} 0 & 0 \\ 0 & \beta_4 \end{bmatrix}$, $\Delta V = M_b \beta_3 q_1 \sin(q_2)$ and $\beta_3$ and $\beta_4$ are the unknown part of the gravitational constant, $g$, and the moment of inertia of the beam, $J$, respectively.

*4.2. Controller Design*

According to the method in [2], $M_d$ is adopted as

$$
M_d = \begin{bmatrix} R\sqrt{\frac{2}{c_3}} & \frac{R^2}{c_3} \\ \frac{R^2}{c_3} & \sqrt{\frac{2}{c_3^3}} R^3 \end{bmatrix}.
\tag{36}
$$

After substituting $M_d$ into (28), $j$ is calculated as

$$
j = \frac{M_b R^3 q_1 \left( 2R\sqrt{2c_3} p_1 p_2 - R^2 p_1^2 - 2c_3 p_2^2 \right)}{c_2^2 \left( \sqrt{2c_3} p_2 - R p_1 \right)}.
\tag{37}
$$

The potential energy is solved by substituting $M$ and $M_d$ into PDE (29), which can be represented as

$$
\sqrt{\frac{2R^2}{c_3}} \frac{\partial V_d}{\partial q_1} + \frac{\partial V_d}{\partial q_2} = \frac{c_2}{R^2} M_b g \sin(q_2).
\tag{38}
$$

Solving (38) for $V_d$ results in

$$
V_d = \frac{c_2 M_b g [1 - \cos(q_2)]}{R^2} + \frac{k_p w^2}{2},
$$

where $w = \left( q_2 - \frac{\sqrt{c_2}}{\sqrt{2M_b}R} \arcsin h \left( \frac{\sqrt{M_b} q_1}{\sqrt{J + J_b}} \right) \right)$ and $k_p$ is the controller parameter.

By substituting $M_d$, $j$ and $H_d^*$ into (22), with some straightforward calculations, the energy shaping term, $u_{es}^*$, is expressed as

$$
u_{es}^* = \frac{R M_b q_1 \left( -\frac{R^2 p_1^2}{\sqrt{c_3}} + \sqrt{2} R p_1 p_2 + \sqrt{c_3} p_2^2 \right)}{\sqrt{2} c_1 c_2} + \varphi(q),
\tag{39}
$$

where

$$\varphi(q) = M_b g q_1 \cos(q_2) - \sqrt{\frac{2}{c_3}} R M_b g \sin(q_2) - \frac{k_p R^3 w}{c_2 \sqrt{2c_3}}.$$

Furthermore, from (23), the damping injection term, $u_{di}^*$, is determined as

$$
\begin{aligned}
u_{di}^* &= k_v \left( \frac{c_3 p_1}{R^2} - \frac{\sqrt{2c_3^3} p_2}{R^3} \right) \\
&\quad + \hat{\beta}_4 (\ddot{q}_2 + \dot{q}_2) + M_b \hat{\beta}_3 q_1 \cos(q_2) + \hat{\beta}_2 \dot{q}_2,
\end{aligned}
\tag{40}
$$

where $k_v$ is the damping injection gain. Moreover, the adaptive law from (21) is constructed as

$$
\begin{aligned}
\dot{\hat{\beta}} &= -\gamma \Phi M_d^{-1} p \\
&= \begin{bmatrix}
-\gamma_1 \left( \frac{\sqrt{2c_3} p_1}{R} - \frac{c_3 p_2}{R^2} \right) \dot{q}_1 \\
-\gamma_2 \left( -\frac{c_3 p_1}{R^2} + \frac{\sqrt{2c_3^3} p_2}{R^3} \right) \dot{q}_2 \\
-\frac{\gamma_3 M_b}{R} \left( 2^{\frac{1}{2}} c_3^{\frac{1}{2}} p_1 \sin(q_2) - \frac{c_3 q_1 p_1 \cos(q_2)}{R} - \frac{c_3 p_2 \sin(q_2)}{R} + \frac{2^{\frac{1}{2}} c_3^{\frac{3}{2}} q_1 p_2 \cos(q_2)}{R^2} \right) \\
-\gamma_4 \left( -\frac{c_3 p_1}{R^2} + \frac{\sqrt{2c_3^3} p_2}{R^3} \right) (\ddot{q}_2 + \dot{q}_2)
\end{bmatrix}
\end{aligned}
\tag{41}
$$

### 4.3. Stability Analysis

In this example, LaSalle's invariance principle is applied to prove the asymptotic stability of the closed-loop system. Under the control of (20) and (21), the state equations of the ball and beam system can be described as

$$
\begin{aligned}
\dot{q}_1 &= \frac{R^2}{c_2} p_1, \\
\dot{q}_2 &= \frac{1}{c_1} p_2, \\
\dot{p}_1 &= -M_b g \sin(q_2) + M_b q_1 \dot{q}_2^2 - \beta_1 \dot{q}_1, \\
&= -M_b g \sin(q_2) + \frac{M_b}{c_1^2} q_1 p_2^2 - \frac{R^2}{c_2} \beta_1 p_1, \\
\dot{p}_2 &= -M_b g q_1 \cos(q_2) - \beta_2 \dot{q}_2 + u^* \\
&= -\frac{\beta_2}{c_1} p_2 + \frac{R M_b q_1 \left( -\frac{R^2 p_1^2}{\sqrt{c_3}} + \sqrt{2} R p_1 p_2 + \sqrt{c_3} p_2^2 \right)}{\sqrt{2} c_1 c_2} \\
&\quad - \sqrt{\frac{2}{c_3}} R M_b g \sin(q_2) - \frac{k_p R^3 w}{c_2 \sqrt{2c_3}} + k_v \left( \frac{c_3 p_1}{R^2} - \frac{\sqrt{2c_3^3} p_2}{R^3} \right) \\
&\quad + \hat{\beta}_4 (\ddot{q}_2 + \dot{q}_2) + M_b \hat{\beta}_3 q_1 \cos(q_2) + \hat{\beta}_2 \dot{q}_2.
\end{aligned}
\tag{42}
$$

and the output equation is

$$
\begin{aligned}
y_d^* &= G^T \nabla_p H_d^* \\
&= \frac{\sqrt{2c_3^3} p_2}{R^3} - \frac{c_3 p_1}{R^2}.
\end{aligned}
\tag{43}
$$

Restricted by manifold $y_d^* \equiv 0$, $\forall t$, the trajectories of the system (42) are analyzed as follows. It follows from $y_d^* = 0$ that

$$p_2 = \sqrt{\frac{1}{2c_3}} R p_1. \tag{44}$$

According to $\dot{y}_d^* = 0$, one obtains

$$
\begin{aligned}
0 = \ & -\frac{c_2 M_b g \sin(q_2)}{R^2} - w - \frac{M_b q_1 p_1^2}{c_1} + \beta_1 p_1 \\
& -\frac{c_3 \tilde{\beta}_2 p_1}{R^2} + \frac{c_2 \sqrt{2c_3} M_b \hat{\beta}_3 q_1 \cos(q_2)}{R^3} \\
& + \frac{\hat{\beta}_4}{c_1} \left( -\frac{M_b q_1 p_1^2}{2c_1} - \frac{c_2 M_b g \sin(q_2)}{R^2} - \beta_1 p_1 + \frac{c_2 p_1}{R^2} \right).
\end{aligned} \tag{45}
$$

From Equation (44), the ball and beam system can be reduced to the following system

$$\dot{q}_1 = \frac{R^2}{c_2} p_1, \tag{46}$$

$$\dot{p}_1 = -M_b g \sin(q_2) + \frac{M_b R^2 q_1 p_1^2}{2c_1 c_2} - \frac{R^2 \beta_1 p_1}{c_2}, \tag{47}$$

$$\dot{q}_2 = \frac{R}{\sqrt{2c_1 c_2}} p_1. \tag{48}$$

According to (46) and (48), it can be easily verified that

$$
\begin{aligned}
\frac{dw}{dt} &= \frac{d}{dt} \left( q_2 - \frac{\sqrt{c_2}}{\sqrt{2M_b} R} \operatorname{arcsin} h \left( \frac{\sqrt{M_b} q_1}{\sqrt{J + J_b}} \right) \right) \\
&= \frac{R}{\sqrt{2c_1 c_2}} p_1 - \frac{\sqrt{c_3}}{\sqrt{2} R} \dot{q}_1 \\
&= 0,
\end{aligned}
$$

which means that

$$q_2(t) - \frac{\sqrt{c_2}}{\sqrt{2M_b} R} \operatorname{arcsin} h \left( \frac{\sqrt{M_b} q_1(t)}{\sqrt{J + J_b}} \right) = 2\delta, \ \forall t. \tag{49}$$

It is known that the origin is an equilibrium point, which requires $\delta = 0$. According to (49),

$$q_2 = \frac{\sqrt{c_2}}{\sqrt{2M} R} \operatorname{arcsin} h \left( \frac{\sqrt{M} q_1(t)}{\sqrt{J + J_b}} \right). \tag{50}$$

After a series of simplifications, the final second-order system is described as:

$$\dot{q}_2 = \frac{R}{\sqrt{2c_1 c_2}} p_1, \tag{51}$$

$$\dot{p}_1 = -M_b g \sin(q_2) + \frac{M_b R^2 q_1 p_1^2}{2c_1 c_2} - \frac{R^2 \beta_1 p_1}{c_2}. \tag{52}$$

**Assumption 1.** *Assume that the following condition is true.*

$$(J + J_b)\beta_1 - \frac{c_2}{R^2} \tilde{\beta}_2 - \hat{\beta}_4 \beta_1 + \frac{c_2}{R^2} \hat{\beta}_4 \neq 0.$$

**Lemma 1.** *With Assumption 1, if the trajectories of (50)–(52) are confined to $y_d^* = 0$ and $\dot{y}_d^* = 0$, that is, (44) and (45), then $q_2(t) \geq 0$, $\forall t$ or $q_2(t) \leq 0$, $\forall t$.*

**Proof of Lemma 1.** Assume $q_2(t^*) = 0$ at $t^*$. It can be obtained from (50) that $q_1(t^*) = 0$. Substituting $(q_1, q_2) = (0, 0)$ into (45) leads to

$$\left( (J + J_b)\beta_1 - \frac{c_2}{R^2}\tilde{\beta}_2 - \hat{\beta}_4\beta_1 + \frac{c_2}{R^2}\hat{\beta}_4 \right) p_1 = 0. \tag{53}$$

According to Assumption 1, (53) has only one real solution, $p_1 = 0$, which implies that $p_2 = 0$. Since $q_1(t^*) = 0$ and $q_2(t^*) = 0$, the fact that $\dot{p}_1(t^*) = 0$ can be obtained from (52). Because $p_1(t^*) = 0$ and $\dot{p}_1(t^*) = 0$, $\ddot{p}_1(t^*) = 0$ because

$$
\begin{aligned}
\ddot{p}_1(t^*) &= -M_b g \cos(q_2)\dot{q}_2(t^*) \\
&\quad + \frac{\left( M_b R^2 \dot{q}_1(t^*) p_1^2 + 2M_b R^2 q_1 p_1 \right)(2c_1 c_2) - M_b R^2 q_1 p_1^2 2M_b q_1 \dot{q}_1(t^*) c_2}{(2c_1 c_2)^2}
\end{aligned}
$$

$$-\frac{R^2 \beta_1 \dot{p}_1(t^*)}{c_2} = 0$$

together with (46). As a result, $p_1(t) = 0$ for $t > t^*$, which implies that $p_2(t) = 0$ for $t > t^*$ due to (44). Since $q_2(t^*) = 0$ and $\dot{q}_2(t) = 0$ for $t > t^*$, because $p_1(t) = 0$ and (51), $q_2(t) = 0$ for $t > t^*$ because $\ddot{q}_2(t^*) = \frac{R\dot{p}_1(t^*)\sqrt{2c_1 c_2} - 4RM_b p_1 q_1 \dot{q}_1(t^*) c_2}{2c_1 c_2} = 0$. With (50), it follows from $q_2(t) = 0$ that $q_1(t) = 0$ for $t > t^*$. Finally, it can be concluded that if $q_2(t^*) = 0$, the system will stay at the origin for $t > t^*$. $\square$

Next, the final second-order system, (51) and (52), is linearized at the origin to obtain

$$
\begin{aligned}
\dot{q}_2 &= \frac{R}{\sqrt{2c_2(J + J_b)}} p_1 \\
\dot{p}_1 &= -M_b g q_2 - \frac{R^2}{c_2}\beta_1 p_1.
\end{aligned}
\tag{54}
$$

The system matrix, $A$, can be obtained from (54) as follows:

$$
A = \begin{bmatrix} 0 & \frac{R}{\sqrt{2c_2(J + J_b)}} \\ -M_b g & -\frac{R^2}{c_2}\beta_1 \end{bmatrix},
$$

and the eigenvalues of matrix $A$ are calculated as

$$
\lambda_1 = -\frac{1}{2c_2}\left( R^2\beta_1 - \sqrt{R^4\beta_1^2 - \frac{2\sqrt{2c_2^3}RM_b g}{\sqrt{J + J_b}}} \right),
$$

$$
\lambda_2 = -\frac{1}{2c_2}\left( R^2\beta_1 + \sqrt{R^4\beta_1^2 - \frac{2\sqrt{2c_2^3}RM_b g}{\sqrt{J + J_b}}} \right).
$$

Whether $R^4\beta_1^2 - \frac{2\sqrt{2c_2^3}RM_b g}{\sqrt{J + J_b}} > 0$ or $R^4\beta_1^2 - \frac{2\sqrt{2c_2^3}RM_b g}{\sqrt{J + J_b}} < 0$, the eigenvalues have a negative real part. So the linearized system (54) is asymptotically stable. Furthermore, the system in (51) and (52) is locally asymptotically stable. Since $q_2 \in (-180°, 180°)$, it can be easily verified that the origin is the only equilibrium point, so $q_2 \to 0$. According to the proof of Lemma 1, $(q, p) = (0, 0)$ can be deduced. Hence, the conclusion that $\lim p = 0$ is proved. The equilibrium point $(q_*, 0) = (0, 0)$ is locally asymptotically stable.

*4.4. Numerical Simulation Results*

In this section, the proposed controller is simulated and compared with the article in [33] under different initial conditions and controller parameters. Matlab software is used to implement this numerical simulation. The legends "Proposed" and "Backstepping" in Figures 2–13 represent the proposed controller and the backstepping method from [33]. The system parameters used for the simulation are $M = 0.05$ kg, $R = 0.01$ m, $J = 0.02$ kg·m$^2$, $J_b = 2 \times 10^{-6}$ kg·m$^2$ and $g = 9.81$ m/s$^2$, which are taken from [28]. The simulation results are shown in Figures 2–13.

Case 1: The initial conditions are chosen as $[q, \dot{q}] = [0, 20°, 0, 0]$ and $\left[\hat{\beta}_1, \hat{\beta}_2, \hat{\beta}_3, \hat{\beta}_4\right] = [0.02, 0, 0, 0]$. The controller parameters are set to $\gamma_1 = 0.001$, $\gamma_2 = 0.1$, $\gamma_3 = 0.5$, $\gamma_4 = 0.001$, $k_p = 1$ and $k_v = 1$. Figures 2–7 show the results under Case 1. It can be obviously seen from Figure 2 that using the proposed adaptive controller ((20) and (41)), the ball reaches the expected position in about 5 s. However, under the backstepping method, the ball gradually stabilizes to the expected position after 50 s. As shown in Figure 3, the beam keeps swinging in a range of $(-16°, 20°)$ from 0 to 0.5 s with the backstepping method. However, under the proposed adaptive controller, the beam reaches $-5°$ in 0.2 s and then gradually converges to the equilibrium position. It can be clearly seen from Figures 4 and 5 that under the action of the proposed adaptive controller, the acceleration of the ball and the angular acceleration of the beam are obviously smaller than under the backstepping method. The acceleration of the ball reaches a maximum value of 1 m/s$^2$ in 0.2 s, and then decays to 0 gradually. In addition, it can be observed from Figure 6 that the control signal of the proposed adaptive controller is much smaller and settles down faster than for the backstepping method. As depicted in Figure 7, the estimated values of parameters are all bounded.

Case 2: The initial conditions are selected as $[q, \dot{q}] = [1m, 0°, 0, 0]$ and $\left[\hat{\beta}_1, \hat{\beta}_2, \hat{\beta}_3, \hat{\beta}_4\right] = [0.02, 0, 0, 0]$. The controller parameters are given as $\gamma_1 = 0.001$, $\gamma_2 = 0.1$, $\gamma_3 = 0.5$, $\gamma_4 = 0.001$, $k_p = 0.8$ and $k_v = 30$. From Figure 8, it can be seen that the swing amplitude of the ball is very small and reaches equilibrium position in about 5 s under the action of the proposed adaptive controller, while under the backstepping method, the ball oscillates more from 0 to 10 s. The angle of the beam reaches a maximum value of 45° at 0.5 s then decreases sharply and stabilizes at the equilibrium position in Figure 9. Using the backstepping method, the beam swings back and forth between $-15°$ and 31° with a large amplitude. From Figures 10 and 11, the acceleration of the ball and the angular acceleration of the beam are significantly smaller than for the backstepping method, and the proposed adaptive controller can better control the position of the ball and the swing angle of the beam. Compared with the backstepping method, the vibration amplitude of the control signal is significantly smaller under the application of the proposed adaptive controller, as shown in Figure 12. It can be seen from Figure 13 that the estimated parameters are bounded.

**Remark 1.** *It is worth noting that the acceleration, $\ddot{q}_2$, is required to implement the proposed controller when there is an unknown term, $\Delta M$, in the inertial matrix, $M$, which limits the applications of the proposed controller. The drawback of the proposed controller design method is that it is more complicated compared with the traditional linear controller design methods, such as pole placement, linear quadratic regulator, proportional integral derivative, etc., and the nonlinear controller design methods, such as backstepping, sliding mode control, approximate linearization and so on.*

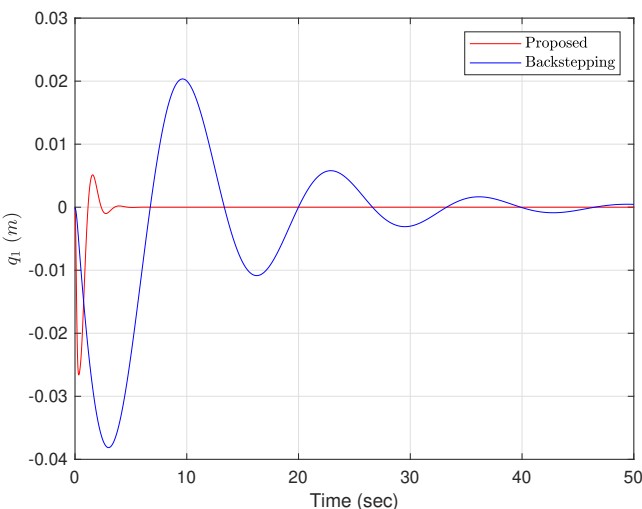

**Figure 2.** The position of the ball. (Case 1).

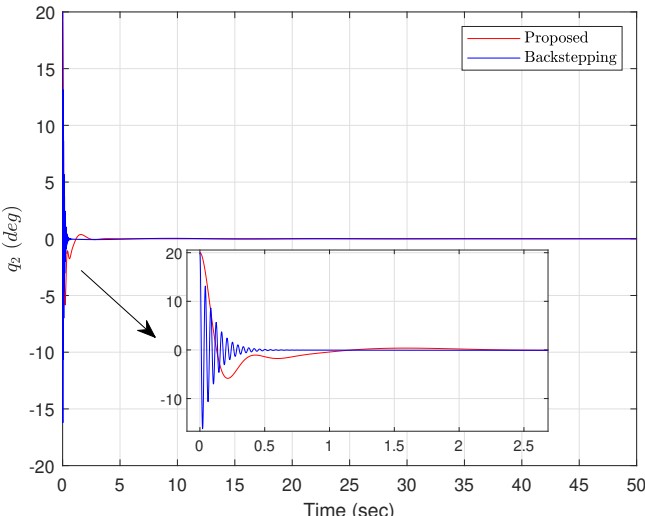

**Figure 3.** The angle of the bar. (Case 1).

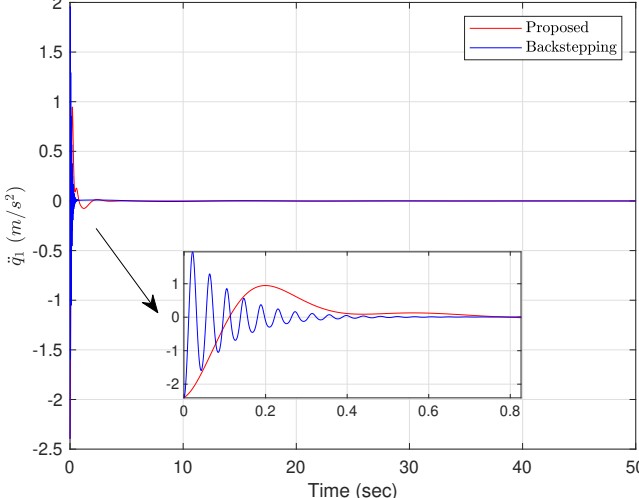

**Figure 4.** The acceleration of the ball. (Case 1).

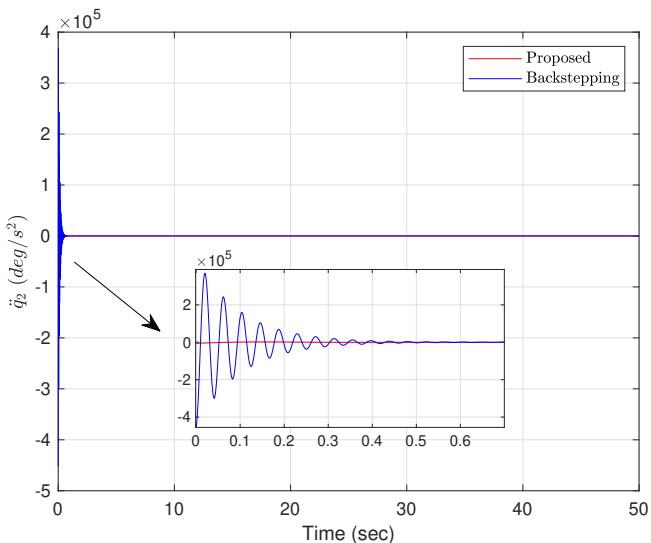

**Figure 5.** The angular acceleration of the beam. (Case 1).

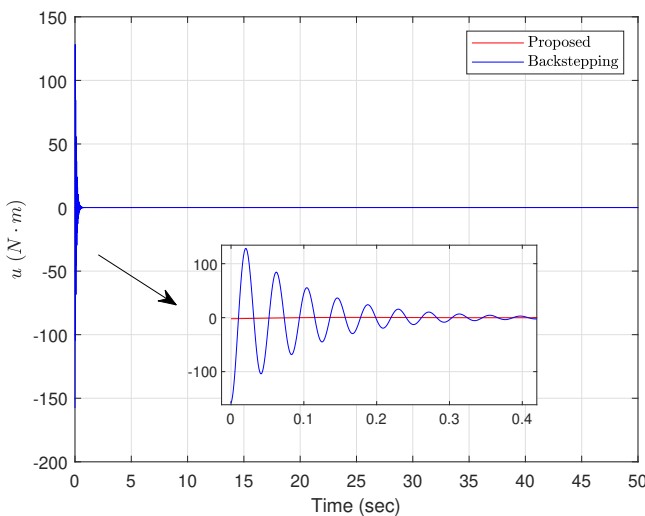

**Figure 6.** The control signal. (Case 1).

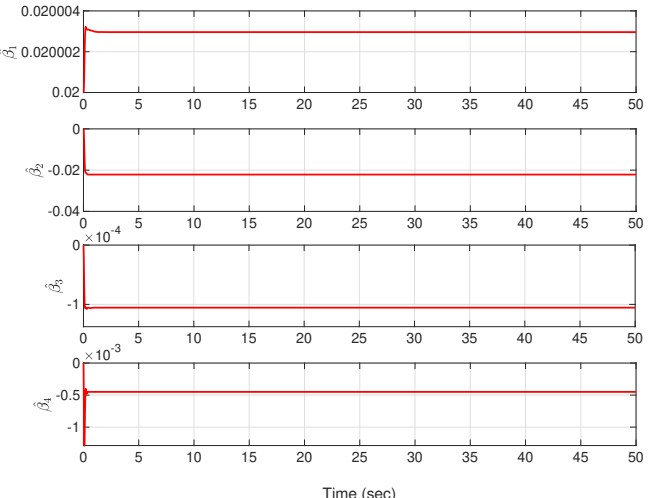

**Figure 7.** The estimated parameters. (Case 1).

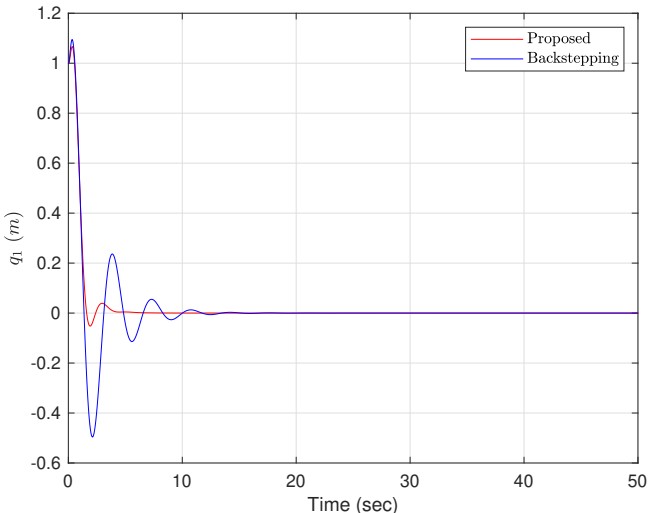

**Figure 8.** The position of the ball. (Case 2).

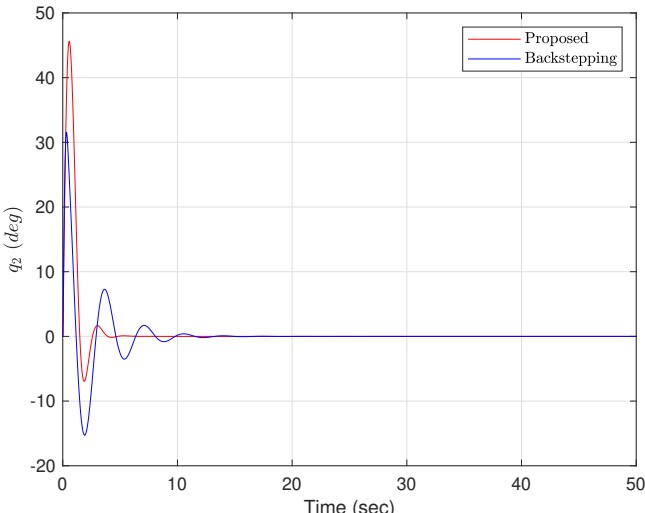

**Figure 9.** The angle of the bar. (Case 2).

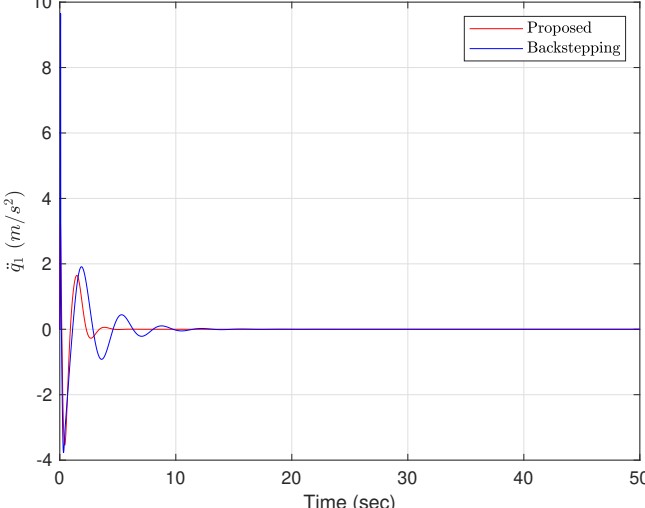

**Figure 10.** The acceleration of the ball. (Case 2).

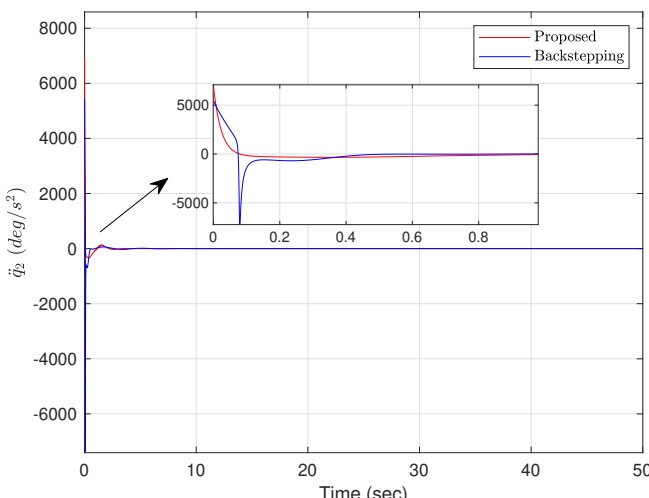

**Figure 11.** The angular acceleration of the beam. (Case 2).

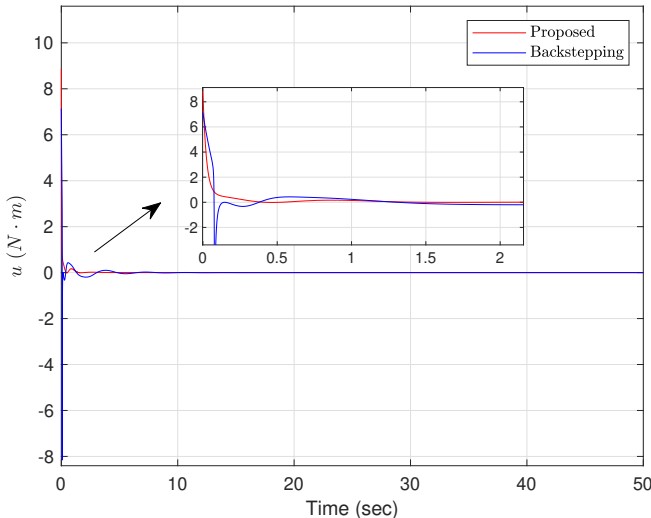

**Figure 12.** The control signal. (Case 2).

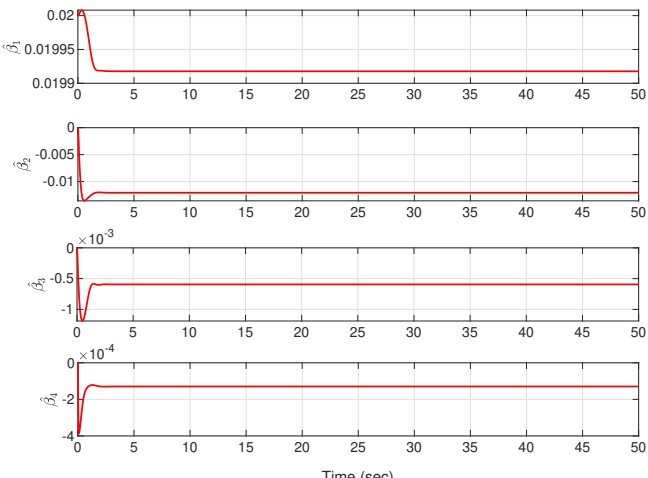

**Figure 13.** The estimated parameters. (Case 2).

## 5. Conclusions

In this paper, an adaptive control law was designed for a class of underactuated mechanical systems with matched and unmatched uncertainties. With this controller,

the locally asymptotic stability of the underactuated mechanical system is ensured under uncertainties. The estimate values of the unknown terms are placed in the damping injection controller $u_{di}$, which simplifies the design of the controller. In order to verify the effectiveness of the proposed controller, it is applied to the ball and beam system. The locally asymptotic stability of the ball and beam system is proved by using LaSalle's invariance principle and approximate linearization. The numerical simulation results show the effectiveness of the control strategy. The proposed adaptive controller can better control the position of the ball and the swing angle of the beam than the backstepping method. Compared with other methods, the proposed adaptive controller is more complicated. Future work will include considering external disturbances and more uncertainties, making this method more general and trying to apply it to underactuated systems such as bridge cranes.

**Author Contributions:** Conceptualization, X.L. and H.S.; methodology, X.L. and H.S.; software, H.S., X.G. and N.L.; validation, H.S., F.Z. and L.S.; writing—original draft preparation, H.S.; writing—review and editing, X.L., C.L. and N.L. All authors have read and agreed to the published version of the manuscript.

**Funding:** This work is financially sponsored by the Innovation Team of Jinan (202228039), The science and Technology SMEs Innovation Ability Improvement Project of Shandong Province (2023TSGC0227, 2023TSGC0100), the Natural Science Foundation of Shandong Province (ZR202110110003).

**Data Availability Statement:** Not applicable.

**Acknowledgments:** The authors acknowledge Xinpeng Guo, Fei Zheng and Lijun Sun for their assistance in this research. With their help, we completed this research together.

**Conflicts of Interest:** The authors declare no conflict of interest.

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
