# Peer review of "An Adaptive Controller Based on Interconnection and Damping Assignment Passivity-Based Control for Underactuated Mechanical Systems: Application to the Ball and Beam System"

_actuators, doi:10.3390/act12110408_

Round 1

Reviewer 1 Report

Comments and Suggestions for Authors

The reviewed manuscript concerns the use of an adaptive controller based on IDA-PBC for underactuated ball and beam system.

The presented approach is interesting and worth publishing. Several editing and linguistic errors were noticed in the text, which do not make reading difficult and will certainly be corrected by the editors. 

However, I suggest introducing some additional information, because in this form I have a few doubts:

1) I think that in the introduction it is worth referring (at least briefly) to current methods of modeling an underactuated ball and beam system and methods of controlling it. This will highlight the novelty of the work and demonstrate that the proposed concept is a new contribution to the development of a device of this class.

2) There is an incorrect reference "[?]"on line 175

3) Perhaps I didn't read it correctly or didn't understand it, but I wonder why the system of equations 32 includes such a non-standard form of dynamic behavior and not the standard equations of motion? 

4) Similarly the expression that the potential energy function V (34) is attained from (32) is incomprehensible. And why it does not take into account the ball's mass? Should this be interpreted as meaning that formula 33 is the initial assumption?

5) Although the ball-beam angle is very well known, I think it is worth adding a simplified model showing the adopted generalized coordinates.

6) As for the numerical simulations, a non-rational assumption was made for case 1. With the beam tilted more than 90 degrees, the ball should fall off the beam - there will definitely be no contact. In the work with which the results are compared, the beam was vertical.

7) I consider the conclusions to be very optimistic. No limits were given, especially when it comes to implementation possibilities. With such high angular accelerations of the beam, it will undoubtedly be difficult to control the position of the ball, because it will be thrown up by the movement of the beam.

Comments on the Quality of English Language

The work should be checked for punctuation. I included additional information in one of the comments.

Reviewer 2 Report

Comments and Suggestions for Authors

This paper presents an adaptative based on interconnection and damping assignment and passivity-based controller for underactuated mechanical systems. Specifically, it is applied to stabilizing a ball and beam system with matched and unmatched uncertainties. The uncertainties considered are unknown frictions and terms of potential and kinetic energy. The locally asymptotic stability is demonstrated using LaSalle's invariance principle and approximate linearization. The paper is well written and argued, however it can be improved with:

* The abstract is repetitive about the uncertainties considered. This is, two times, the unknown friction coefficients and unknown terms of kinetic and potential energy. Please avoid this repetition. 

* To improve the literature review. I suggest including more recent references about IDA-PBC that also deal with noise and disturbance, e.g., filtered observer-based IDA-PBC control for trajectory tracking of a quadrotor, IEEE Access; Trajectory tracking control for unmanned surface vessel with input saturation and disturbances via robust state error IDA-PBC approach, j. of the frankling institute; Passivity-based control laws for an unmanned powered parachute aircraft, asian journal of control. 

* The main results are in section 3, where the controller is designed, and the stability is analyzed according to the description contained in the firsts lines of this section. However, the section is very brief. In order to make the contribution clearer, put the main result of this paper as a Theorem, where you state the stability properties of the control law and provide the corresponding proof. 

* Include remarks about the limitations of the proposed approach. 

*  Section 4 contains the application of the ball and beam. Please note that the implementation of this controller is to validate the theoretical developments and should not be the core of the paper. Then, I would suggest analyzing the stability in the most general form (section 3) and not just in the particular case of the ball and beam. 

* Please allow a blank space between the variable name and the units in the plots. The same in the figure captions about what is written in parentheses.

* Please be specific on what you call "The existing IDA-PBC"; use a reference. Comparisons must be made with recent approaches. 

* Unify the use of radians/degrees in the document. You are using radians in some parts and degrees in other parts of the document, e.g. in Figures 2,6 are in degrees and in page 10, line 215, you use radians. 

* Expand the conclusions. The conclusion section reads as a summary. Add conclusions. Add future proposed work. 

Comments on the Quality of English Language

Minor editing required

Round 2

Reviewer 1 Report

Comments and Suggestions for Authors

Dear Authors,

After reviewing the revised version of the manuscript, I think it can be published in the form presented.

Reviewer 2 Report

Comments and Suggestions for Authors

I am happy with the revised manuscript. 

Comments on the Quality of English Language

Minor corrections required